# Evaluation of Fluoroquinolone Resistance in Clinical Avian Pathogenic *Escherichia coli* Isolates from Flanders (Belgium)

**DOI:** 10.3390/antibiotics9110800

**Published:** 2020-11-12

**Authors:** Robin Temmerman, An Garmyn, Gunther Antonissen, Gerty Vanantwerpen, Mia Vanrobaeys, Freddy Haesebrouck, Mathias Devreese

**Affiliations:** 1Department of Pharmacology, Toxicology and Biochemistry, Faculty of Veterinary Medicine, Ghent University, Salisburylaan 133, 9820 Merelbeke, Belgium; robin.temmerman@ugent.be (R.T.); gunther.antonissen@ugent.be (G.A.); 2Department of Pathology, Bacteriology and Avian Diseases, Faculty of Veterinary Medicine, Ghent University, Salisburylaan 133, 9820 Merelbeke, Belgium; an.garmyn@ugent.be (A.G.); freddy.haesebrouck@ugent.be (F.H.); 3Animal Health Care Flanders (DGZ), Industrielaan 29, 8820 Torhout, Belgium; gerty.vanantwerpen@poulpharm.be (G.V.); mia.vanrobaeys@dgz.be (M.V.)

**Keywords:** antimicrobial resistance, APEC, enrofloxacin, MIC, MPC

## Abstract

Fluoroquinolones are frequently used antimicrobials for the treatment of avian pathogenic *Escherichia coli* (APEC) infections. However, rapid development and selection of resistance to this class of antimicrobial drugs is a significant problem. The aim of this study was to investigate the occurrence and mechanisms of antimicrobial resistance against enrofloxacin (ENRO) in APEC strains in Flanders, Belgium. One hundred and twenty-five APEC strains from broilers with clinical colibacillosis were collected in Flanders from November 2017 to June 2018. The minimum inhibitory concentration (MIC) of all strains and the mutant prevention concentration (MPC) of a sample of sensitive isolates were determined using a commercial gradient strip test and via the agar dilution method, respectively. Non-wild type (NWT) isolates were further characterized using polymerase chain reaction (PCR), gel electrophoresis and gene sequencing. Forty percent of the APEC strains were NWT according to the epidemiological cut-off (ECOFF) measure (MIC > 0.125 μg/mL). With respect to clinical breakpoints, 21% were clinically intermediate (0.5 ≤ MIC ≤ 1 μg/mL) and 10% were clinically resistant (MIC ≥ 2). The MPC values of the tested strains ranged from 0.064 to 1 μg/mL, resulting in MPC/MIC ratios varying from 4 to 32. The majority (92%) of the NWT strains carried one or two mutations in gyrA. Less than a quarter (22%) manifested amino acid substitutions in the topoisomerase IV parC subunit. Only three of the NWT strains carried a mutation in parE. Plasmid mediated quinolone resistance (PMQR) associated genes were detected in 18% of the NWT strains. In contrast to the relatively large number of NWT strains, only a small percentage of APEC isolates was considered clinically resistant. The most common MPC value for sensitive strains was 0.125 μg/mL. Some isolates showed higher values, producing wide mutant selection windows (MSW). Chromosomal mutations in DNA gyrase and topoisomerase IV were confirmed as the main source of decreased antimicrobial fluoroquinolone susceptibility, de-emphasizing the role of PMQR mechanisms.

## 1. Introduction

*Escherichia coli* (*E. coli*) is a normal constituent of the residential microbiota present in the intestinal tract of animals, including birds [1]. Most of these bacteria play an important beneficial role in gut health and homeostasis [2]. In contrast, a minority (10–15%) of the *E. coli* intestinal population in poultry has the ability to cause disease [3,4]. These genotypically heterogeneous strains carry a variety of virulence factors and are considered to be members of extraintestinal pathogenic *E. coli* (ExPEC) [5,6,7]. The specific avian pathotype, able to cause disease in birds, is aptly named avian pathogenic *E. coli* (APEC). The APEC group contains different serotypes. The O1, O2, O8, O18 and O78 serotypes are the most frequently detected in clinical cases in poultry [8]. Colibacillosis refers to any localized or systemic infection caused by APEC [9,10]. In broiler chickens, colibacillosis is associated with different disease entities including colisepticemia, coligranuloma (Hjarre’s disease), airsacculitis (chronic respiratory disease, CRD), swollen-head syndrome (SHS), coliform cellulitis and omphalitis [10].

Prevention and control of colibacillosis depends on effective biosecurity measures and vaccination [1]. However, as a consequence of the notorious genomic diversity of these bacteria, vaccines often fail to provide adequate protection [4,7,11,12]. Therefore, colibacillosis is still a worldwide problem, which mostly relies on curative and metaphylactic antimicrobial therapy [13]. Enrofloxacin (ENRO), a second generation member of the fluoroquinolone chemotherapeutics family and solely used in veterinary medicine, is regularly administered via the drinking water to poultry [14,15,16]. Regrettably, because of widespread (mis)use, many *E. coli* isolates show high levels of fluoroquinolone resistance [14].

For the purpose of limiting the emergence of resistant subpopulations of bacteria during treatment, drug plasma concentrations should exceed the mutant prevention concentration (MPC) threshold. The MPC, first conceptualized in 1999 [17], has been defined as the lowest concentration of an antimicrobial drug that inhibits the emergence of mutant bacteria from an inoculum of 10^10^ colony forming units (cfu) [18,19] and is determined via agar dilution. The MPC concept fits in the mutant selection window (MSW) hypothesis, which states that in the concentration range between the minimum inhibitory concentration (MIC) and the MPC (i.e., the MSW), resistant bacteria will be selectively enriched [18].

Vanni and colleagues [20] reviewed the fluoroquinolone resistance rates of *E. coli* isolated from broilers around the globe (Europa, USA and Asia). The prevalence of ENRO resistance varied greatly across countries: from 16% in the USA to 90% in China. Such geographical variability likely reflects differences in local antimicrobial usage and methodological differences between the studies [13]. In conjunction with high reported levels of resistance against other antimicrobial classes such as beta-lactams, tetracyclines, aminoglycosides and potentiated sulphonamides, resistance against the fluoroquinolone class would severely limit therapeutic options [20,21,22]. Moreover, veterinary use of fluoroquinolones is heavily scrutinized because of association with fluoroquinolone resistant food-borne zoonotic bacteria such as *E. coli*, *Salmonella* and *Campylobacter* [20,23].

Resistance against fluoroquinolones in Gram-negative bacteria generally develops in a stepwise manner through accumulation of specific, conservative point mutations in primarily the genes encoding for DNA gyrase (*gyrA* and *gyrB*) and secondary in genes encoding for DNA topoisomerase IV (*parC* and *parE*) [14]. These enzymes play a crucial role in the remodeling of DNA topology and DNA replication [24,25]. The regions of DNA sequences where these target-site mutations rise are termed the quinolone resistance determining regions (QRDR) [26]. Mutations in this regional genetic code, resulting in amino acid substitutions, alter the protein structure of the target and prompt the abatement of fluoroquinolone susceptibility [25].

In contrast to chromosomal mutations, which are mainly transferred vertically, resistance genes located on plasmids can be propagated horizontally between bacteria (horizontal gene transfer, HGT). For the fluoroquinolone class, this phenomenon is described as plasmid-mediated quinolone resistance (PMQR). For many years, it was thought that plasmid-mediated resistance mechanisms against fluoroquinolones were almost non-existent [27]. However, since the first discovery of plasmid-mediated quinolone resistance [28] and subsequent confirmation [29], multiple genes conferring resistance to fluoroquinolones located on plasmids have been described [25,30]. The translated proteins are involved in target protection (*qnr A,B,C,D,E,S*), reducing fluoroquinolone efficacy (*aac(6′)-Ib-cr*) and efflux mechanisms (*oqxAB* and *qepA*) [27,30,31].

The aim of this study was to investigate the occurrence and molecular mechanisms of antimicrobial resistance against ENRO in clinical APEC isolates from Flanders (Belgium). The strains were first serotyped based on the most common occurring APEC antigens associated with colibacillosis in Western Europe (O1, O2 and O78) [32]. Next, the MIC of all strains and MPC values of 20 selected sensitive isolates were determined. Finally, the genetic determinants of fluoroquinolone resistance (QRDR and PMQR) were characterized. More specific, chromosomal (*gyrA*, *parC* and *parE*) and plasmid-encoded resistance genes (*qnrA*, *qnrB*, *qnrS*, *qepA* and *oqxAB*) were evaluated. With the resistance pattern of APEC against ENRO known, a more substantiated, evidence-based treatment can be considered. Additionally, such susceptibility data of target pathogens can be used for antimicrobial dose optimization in order to mitigate the development of antimicrobial resistance, thereby increasing the sustainability of antimicrobial agents in veterinary medicine.

## 2. Results

### 2.1. Serotyping

O78 was the most prevalent determined serotype (19% of the strains). Five and eight percent belonged to the O1 and O2 serotype, respectively. The remaining 68% of the isolated APEC strains were non-identifiable using the three antisera.

### 2.2. MIC Distribution

The results of the quality control bacteria (*E. coli* ATCC 25922) were within the acceptable control ranges in accordance to the Clinical and Laboratory Standards Institute (CLSI) guidelines [33], namely between 0.008 and 0.03 μg/mL. The determined MIC values ranged from 0.008 to more than 32 μg/mL. Using the epidemiological cut-off (ECOFF) as the classification method, 60% of the strains were WT and subsequently the remaining 40% were considered NWT (microbiologically resistant). Based on the clinical breakpoints, 69% of the isolates were considered susceptible, 21% susceptible with increased exposure (intermediate) and 10% resistant. The MIC distribution of the 125 APEC strains is displayed in Figure 1. The distribution expressed a trimodal aspect, with a very prominent peak at the lower (WT) MIC values, a clear middle peak (intermediate strains) and a lightly pronounced prominence containing resistant bacteria.

### 2.3. MPC Distribution of Sensitive APEC Strains

The sampled colonies on the MH agar plates evaluated with MALDI-TOF were all *E. coli* species, indicating no contamination. The obtained MPC results from the sample of sensitive (MIC values of 0.016 and 0.32 μg/mL) APEC strains are listed in Table 1. Figure 2 gives an overview of the distribution of MPC values (60 measurements in total). An MPC of 0.125 μg/mL was most common (76.67%), associated with a MPC/MIC ratio of 4 or 8 (depending on the sensitivity of the strain). A lower MPC value of 0.064 μg/mL occurred in 6.67% of the cases and was only associated with strains with an MIC of 0.016 μg/mL. The frequency of higher MPC values was 10% (0.25 μg/mL), 5% (0.5 μg/mL) and 1.7% (1 μg/mL). These MPC’s were associated with MPC/MIC ratios ranging from 8 to 32.

### 2.4. Antimicrobial Resistance Genes

The amino acid substitutions in the QRDR of *gyrA*, *ParC* and *parE* and the presence of PMQR genes of the NWT strains are reported in Table 2. None of the WT control strains carried a mutation in the QRDR or possessed a plasmid associated resistance gene.

Ninety-two percent of the NWT strains carried one or two mutations in *gyrA*. The majority of these strains (82%) carried a single amino acid substitution at position 83, replacing serine (S) with leucine (L). Such single mutations confer rather low levels of resistance, as indicated by MIC values of 0.25–1 μg/mL. The remaining strains (18%) carried a second mutation at position 87, resulting in a switch from aspartic acid (D) to asparagine (N). The strains with 2 *gyrA* mutations always possessed additional amino acid substitutions in *parC* and in some cases in *parE*.

With respect to the chromosomal mutations in the topoisomerase IV subunit, 22% of the strains with an MIC value above the ECOFF carried a single mutation in *parC* with occasionally an additional mutation in *parE*. Mutations in *parC* manifested at position 80 as either a swap from serine (S) to arginine (R) or from serine (S) to isoleucine (I). The former occurred less frequently (in 4 of the 11 cases) than the latter (7/11). *ParC* mutations always concurred with one (2/11) or two (9/11) *gyrA* mutations, producing moderate (MIC 1–2 μg/mL) and high (MIC 8–32 μg/mL) resistance levels, respectively.

Only 3 of the NWT strains carried a mutation in *parE*. A serine (S) was replaced with alanine (A) at position 458. This mutation always co-occurs with 2 *gyrA* mutations and one *parC* mutation, resulting in high levels of enrofloxacin resistance (MIC ≥ 32 μg/mL).

PMQR associated genes were detected in 18% of the NWT strains. *QnrS* was most frequently detected (14%), followed by *qnrB* (4%). The other investigated plasmid associated resistance genes, *qnrA*, *qepA* and *oqxAB*, were not detected. Presence of *qnrS* or *qnrB* without additional chromosomal mutations (4/9) was associated with low to moderate decreased susceptibility (MIC 0.25–2 μg/mL). In contrast, when a single point mutation was present in *gyrA* (5/9), higher MIC values were observed (MIC 2–8 μg/mL).

## 3. Discussion

The increasing magnitude of antimicrobial resistance is a major challenge to modern health security and livestock production [34,35,36]. ENRO resistance is a One Health issue, where animals, humans and their shared environment are inextricably intertwined. Next to treatment failure of colibacillosis in chickens, decreased fluoroquinolone susceptibility in APEC also has a dual impact on the human population. First, studies have shown that APEC strains have zoonotic potential [8,37,38,39]. Second, there is the potential of transfer of resistance genes between animals and humans. The environment can play a facilitating role in this process. This dissemination of resistance is predominantly mediated by HGT of plasmids [13,40]. In Flanders (Belgium), there is still a substantial lacuna regarding the prevalence and mechanisms of fluoroquinolone resistance in clinical APEC isolates. Responding to this, the aim of this study was to characterize clinical APEC isolates and their resistance pattern of broiler chickens from this region.

A large number of strains (40%) were considered NWT according to the ECOFF measure, meaning that these bacteria carried one or more resistance elements. In contrast to the relatively high number of NWT strains, only a minority of this group was designated as resistant (R), indicating the presence of a considerable intermediate portion of strains. The shape of the MIC distribution was comparable to the trimodal distribution described in a Chinese study [41]. The main difference between the two distributions was the density of the last part. This component was more prominent for the distribution reported by Sang and colleagues, indicating a proportionally larger amount of resistant strains compared to our collection of APEC strains [42,43].

Next, the MPC values of 20 susceptible, WT isolates were determined. The reason for selecting only susceptible, WT isolates was two-fold. First, since in many parts of the world treatment with fluoroquinolones is only allowed when the pathogen is determined susceptible, mostly susceptible strains are subject to the selective pressure of the antimicrobial treatment. Second, based on the results from the MIC determinations, the WT strains constitute the main part of the APEC population in Flanders. In alignment with previous studies, the MPC values of the tested isolates were fairly stable across the different experiments [44,45,46,47,48]. In 76.67% of the cases, the MPC was 0.125 μg/mL. This corresponded with MPC/MIC ratios of 4 (MIC 0.032 μg/mL) or 8 (MIC 0.016 μg/mL). These results dovetail with other reports of MPC measurements in *E. coli* and APEC strains [14,49].

Approximately 17% of the measurements had higher MPC values. Two strains exhibited a MPC/MIC ratio of 32 (MPC 0.5 and 1 μg/mL, respectively), indicating a large MSW. A wide MSW increases the risk of selective enrichment of resistant subpopulations in a bacterial culture. This inter- and intrastrain variability in MPC values has been reported in other studies [46,48]. Depending whether the mutation occurs early or late during the growth of the bacterial population, there will be many or few resistant bacteria, respectively. This possibly results in higher or lower MPC values. The variation in the frequency of spontaneous mutations in a bacterial population is referred to as Luria–Delbruck fluctuations [48,50].

The results of the QRDR sequencing analysis were largely in agreement with the frequently reported mutations in *E. coli* [20,22,51,52,53]. This was notably the case for *gyrA* and *parC* substitutions. Regarding *gyrA*, two mutations were detected, namely S83L and D87N. The latter occurred only in conjunction with the former. Oppositely, other studies detected the occurrence of single D87N mutations [22,52]. With regards to the *parC* gene, we detected two mutations, namely at codon 80 (S80I and S80R). Mutations in *parC* were always conjointly present with one or two *gyrA* mutations, indicating that these mutations are important in conferring high level fluoroquinolone resistance. This finding is consistent with other research [20,22,52].

Our study detected genetic alterations in the DNA sequence of the *parE* gene of three highly resistant strains (MIC ≥ 32 μg/mL). The point mutation, resulting in the conversion of serine to alanine at position 458 (S458A), has been reported in another study regarding quinolone resistance in extended-spectrum-beta-lactamase-producing *E. coli* [54].

Only a minority (18%) of the NWT APEC strains tested positive for PMQR elements, namely *qnrS* or *qnrB*. The *qnrS* gene was most prevalent (14%), which is consistent with other studies [13,55]. The presence of *qnr* genes was associated with low to moderate levels of decreased susceptibility. The low prevalence of PMQR in the strains with decreased susceptibility corroborates the premise that fluoroquinolone resistance is still primarily induced via chromosomal mutations.

Lastly, this study is subject to some limitations. Firstly, only a limited number of the most prevalent serotypes were investigated. Secondly, since the MPC value is dependent on probability, the MIC value is not a good predictor for the MPC value since there could be substantial variation in the size of the MSW [46]. As indicated by Gianvecchio and colleagues [48], the number of replicates used in our research (3) and in the other mentioned studies might be too few in order to detect even greater stochasticity on the emergence of resistant mutant bacteria. Thirdly and finally, with regards to genotyping, chromosomal mutations in *gyrB* and the presence of other *qnr* genes (*C, D* and *E*) and the variant of the aminoglycoside acetyltransferase, which is responsible for fluoroquinolone resistance *aac(6′)-Ib-cr* were not investigated. However, mutations in *gyrB* are rare and play a minor role in the development of resistance against quinolones in Gram-negative bacteria [55,56]. Likewise, for the non-evaluated PMQR genes, studies have indicated that these plasmid associated genes play a very limited role in fluoroquinolone resistance in bacteria originating from food-producing animals [13,57,58].

## 4. Materials and Methods

### 4.1. APEC Strains

A total of 125 avian pathogenic *E. coli* isolates were included. The isolates were collected in collaboration with Animal Health Care Flanders (Torhout, Belgium) and Sciensano (Brussels, Belgium). The strains originated from different colibacillosis outbreaks that occurred in broiler farms (one strain per farm) located in Flanders (Belgium) from November 2017 until June 2018. The isolated bacteria were identified as *Escherichia coli* using matrix-assisted laser desorption/ionization-time of flight (MALDI-TOF) mass spectrometry. After collection, the bacteria were stored at approximately −70 °C.

### 4.2. Serotyping

Determination of the most common O-antigens associated with colibacillosis, O1, O2 and O78, was performed by slide agglutination using commercially available O-antisera (Ceva Biovac, Beaucouzé, France). Briefly, colonies of bacteria incubated overnight (35 ± 2 °C) on MacConkey agar (Oxoid, Fisher Scientific, Merelbeke, Belgium) were suspended in a droplet of antiserum on a glass slide. A positive result produced agglutination (clumping of cells).

### 4.3. MIC Determination (Gradient Strip Test) and Classification

The MIC was measured via the gradient-strip method (MIC strip test, Liofilchem s.r.l., Roseto degli Abruzzi, Italy). This method has a very high essential and categorical agreement with the gold standard microdilution technique for the specific APEC—ENRO combination [59]. *Escherichia coli* ATCC 25,922 was used as the quality control strain. The procedure was carried out in accordance with previous studies [60,61]. Several colonies (1–5) from the APEC strains, grown overnight on MacConkey agar (35 ± 2 °C), were added to a glass tube containing 3 mL sterile phosphate buffered saline (PBS) and vortexed in order to achieve a 0.5 McFarland inoculum (1.5 × 10^8^ colony forming units (cfu)/mL; ATB 1550 densitometer, Biomerieux, Schaarbeek, Belgium). This suspension was streaked onto Mueller–Hinton (MH) agar plates (BD BBL™, Fisher Scientific, Merelbeke, Belgium). Finally, the MIC test strips were placed at the centre of the plate and incubated (18–24 h at 35 ± 2 °C). The next day, the MIC was determined by examining the intersection between the bacterial growth inhibition zone and the gradient strip. Results were rounded up to the nearest log_2_ dilution, conform results obtained via microbroth or agar dilution.

Considering the MIC results, the strains were categorized via two methods. First, the strains were branched into wild type (WT) and non-wild type (NWT), based on the epidemiological cut off (ECOFF), which is 0.125 μg/mL for ENRO in *E. coli* [62]. The distinction between WT and NWT relies on the absence or presence of an acquired or mutational resistance mechanism to ENRO [63,64].

Second, the strains were classified as susceptible (S, MIC ≤ 0.25 μg/mL), intermediate (I, 0.5 ≤ MIC ≤ 1 μg/mL) or resistant (R, MIC ≥ 2 μg/mL) based on the CLSI clinical breakpoints of fluoroquinolones for *E. coli* from poultry [33]. Recently, the European Committee on Antimicrobial Susceptibility Testing (EUCAST) changed the definition for I from “intermediate” to “susceptible, increased exposure” [65].

### 4.4. MPC Determination (Agar Dilution)

The MPC was determined on 20 randomly selected fully susceptible strains (MIC 0.016 and 0.032 μg/mL). The procedure for MPC evaluation was based on the agar dilution method described in previous studies [14,66]. To summarize, APEC bacteria were cultured overnight (35 ± 2 °C) in 20 mL Mueller Hinton broth (MHB). After incubation, the bacterial suspension was centrifuged (10 min, 2850× *g*, 4 °C) and resuspended in 1 mL MHB in order to yield a concentration of approximately 10^10^ cfu/mL. The inoculum sizes were confirmed through serial 1:10 dilutions in sterile PBS using 96 well plates and plating on MacConkey agar. Next, a series of MH agar plates containing ENRO concentrations that equaled two-fold increases (1×, 2×, 4×, 8× and 16×) of the previously determined MICs of the respective strains were inoculated using a cotton swab. Finally, the plates were incubated for 5 days at 35 ± 2 °C. Each day, the growth of the bacteria was assessed. The lowest drug concentration of the plates where no visible bacterial growth was detected after 3 consecutive days was designated as the MPC. Additionally, some growing colonies on the agar plates were screened for contamination using MALDI-TOF. The MPC experiments were done in triplicate on separate occasions.

### 4.5. Determination of Molecular QRDR and PMQR Using PCR, Gel Electrophoresis and Sequencing

All previously determined NWT strains and 10 control WT strains were selected for genotyping. DNA was extracted from overnight grown bacterial colonies using the PrepMan^®^ Ultra sample preparation reagent and according to the manufacturer’s instructions (Thermo Fisher Scientific).

Primers and protocols used for PCR amplification of the QRDR (*gyrA*, *parC* and *parE*) and the PMQR genes (*qnrA*, *qnrB*, *qnrS*, *qepA* and *oqxAB*) are listed in Table 3. A MasterCycler Gradient EPS-S Thermal Cycler (Eppendorf AG, Hamburg, Germany) was used. The reaction mixture (20 μL in each well of a 96 well plate) consisted of 10 μL of SensiMix™ SYBR^®^ No-ROX master mix (Bioline, London, UK), 0.5 μL of a 20 μM solution of each primer (forward and reverse), 8 μL of HPLC H_2_O and 1 μL of the DNA sample.

Following amplification, the PCR products were evaluated for their presence and quality using agarose gel electrophoresis and UV detection. All QRDR genes should be present, since they constitute the basic genome. Presence of PMQR genes indicated decreased susceptibility against fluoroquinolones.

All obtained amplicons of *gyrA*, *parC* and *parE* were sequenced using the Sanger technique (Eurofins Genomics GmbH, Ebersberg, Germany). The obtained nucleotide and derived amino acid sequences were analyzed and compared with *Escherichia coli* ATCC 25,922 (taxid:1322345) via BioNumerics 7 software (Applied Maths NV, Sint-Martens-Latem, Belgium) and the basic local alignment search tool (BLAST) in order to investigate the presence of point mutations and amino acid substitutions in the QRDR [72].

## 5. Conclusions

In conclusion, these findings illustrate that chromosomal mutations in DNA gyrase and topoisomerase IV are the main source of fluoroquinolone resistance in *E. coli*, de-emphasizing the role of PMQR mechanisms. Despite the relatively large number of NWT strains present in the clinical APEC isolates sample, only a low percentage could be classified as resistant according to the CLSI breakpoint for resistance (MIC ≥ 2 μg/mL). Due to the relatively low level of resistance, ENRO is still a substantiated therapeutic option for the treatment of colibacillosis in Flanders. However, it is critical to use this class of antimicrobial drugs with the utmost prudence in order to mitigate the development of antimicrobial resistance. In Belgium, fluoroquinolone use in the veterinary sector is tightly regulated and administration of this class of antimicrobial drugs is only possible if several conditions are met [73,74]. This has led to a significant decrease in the veterinary use of fluoroquinolones [75]. Next to reducing the overall antimicrobial usage of fluoroquinolones and adherence to the principles of judicious antimicrobial use, other opportunities for sustainable fluoroquinolone use in the veterinary sector include the optimization of currently approved dosage regimens taking into account the current susceptibility pattern of the target pathogens [76].

## Figures and Tables

**Figure 1 antibiotics-09-00800-f001:**
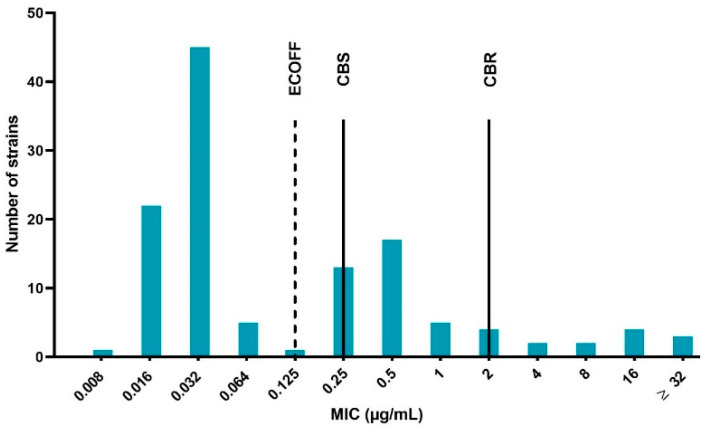
Enrofloxacin minimum inhibitory concentration (MIC) distribution of the 125 avian pathogenic *Escherichia coli* (APEC) strains isolated from broiler farms in Flanders with colibacillosis outbreaks. The dotted line represents the epidemiological cut-off (ECOFF), which is 0.125 μg/mL. Strains with MIC values ≤ ECOFF are labeled wild type (WT) and strains with higher MIC values than the ECOFF as non-wild type (NWT). The full lines marked CBS and CBR indicate the clinical breakpoint for susceptibility (0.25 μg/mL, CBS) and resistance (2 μg/mL, CBR), respectively. Strains with MIC in between CBS and CBR are designated intermediate.

**Figure 2 antibiotics-09-00800-f002:**
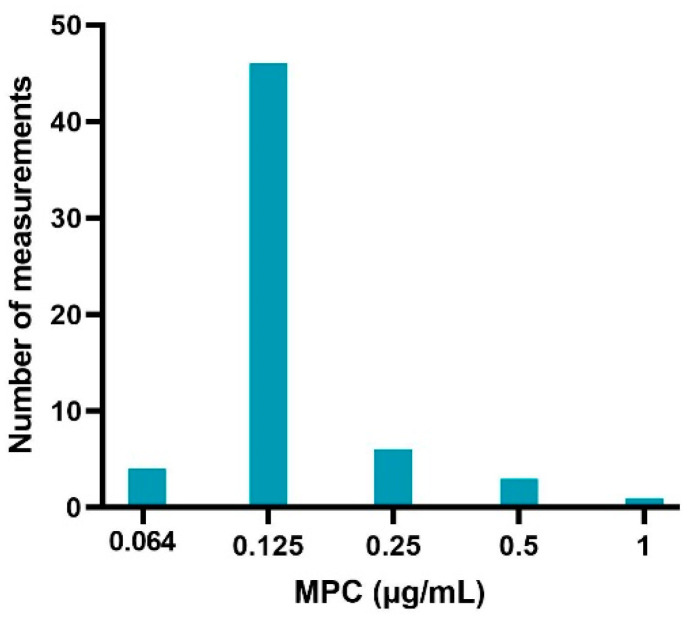
Enrofloxacin MPC distribution of 20 sensitive avian pathogenic *Escherichia coli* (APEC) strains obtained via agar dilution experiments, performed in triplicate (60 measurements in total).

**Table 1 antibiotics-09-00800-t001:** Summary of the mutant prevention concentration (MPC) values of 20 selected sensitive avian pathogenic *Escherichia coli* (APEC) strains obtained via agar dilution experiments performed in triplicate. The width of the mutant selection window (MSW), indicated by the MPC over the MIC ratio, is given between brackets next to the MPC values.

APEC Strain	MIC (μg/mL)	MPC Test 1 (μg/mL)	MPC Test 2 (μg/mL)	MPC Test 3 (μg/mL)
1	0.032	0.125 (4)	0.125 (4)	0.125 (4)
2	0.016	0.064 (4)	0.125 (8)	0.125 (8)
3	0.016	0.125 (8)	0.25 (16)	0.125 (8)
4	0.016	0.125 (8)	0.125 (8)	0.125 (8)
5	0.032	0.125 (4)	1 (32)	0.5 (16)
6	0.016	0.064 (4)	0.125 (8)	0.125 (8)
7	0.032	0.125 (4)	0.125 (4)	0.125 (4)
8	0.032	0.064 (2)	0.25 (8)	0.5 (16)
9	0.016	0.125 (8)	0.125 (8)	0.125 (8)
10	0.032	0.125 (4)	0.125 (4)	0.125 (4)
11	0.016	0.125 (8)	0.125 (8)	0.125 (8)
12	0.032	0.125 (4)	0.125 (4)	0.125 (4)
13	0.032	0.125 (4)	0.25 (8)	0.125 (4)
14	0.032	0.125 (4)	0.25 (8)	0.125 (4)
15	0.032	0.125 (4)	0.125 (4)	0.125 (4)
16	0.032	0.125 (4)	0.125 (4)	0.125 (4)
17	0.032	0.125 (4)	0.125 (4)	0.125 (4)
18	0.016	0.064 (4)	0.125 (8)	0.125 (8)
19	0.032	0.125 (4)	0.25 (8)	0.25 (8)
20	0.016	0.125 (8)	0.125 (8)	0.5 (32)

**Table 2 antibiotics-09-00800-t002:** Overview of the fluoroquinolone resistance mechanisms (QRDR and PMQR) of the different NWT strains (strains with MIC values above the ECOFF).

MIC (μg/mL)	Number of Isolates	QRDR Mutations			PMQR
		*gyrA*	*parC*	*parE*	
0.25	1				*qnrS*
	12	S83L			
0.5	17	S83L			
1	2				*qnrS*
	2	S83L			
	1	S83L	S80R		
2	1				*qnrS*
	2	S83L			*qnrB*
	1	S83L	S80R		
4	2	S83L			*qnrS*
8	1	S83L			*qnrS*
	1	S83L/D87N	S80R		
16	4	S83L/D87N	S80I		
≥32	1	S83L/D87N	S80R	S458A	
	2	S83L/D87N	S80I	S458A	

**Table 3 antibiotics-09-00800-t003:** Overview of the PCR primers used in this study to detect chromosomal (QRDR) and plasmid quinolone resistance (PMQR) genes.

Target	Primer	Sequence (5′ → 3′)	Annealing T (°C)	Size of Product (bp)	Reference
*gyrA*	Gyr A 6-F	CGA CCT TGC GAG AGA AAT	57	625	[67]
	Gyr A 631-R	GTT CCA TCA GCC CTT CAA			
*parC*	Par C 137-F	TGT ATG CGA TGT CTG AAC TG	55	265	[68]
	Par C 401-R	CTC AAT AGC AGC TCG GAA TA			
*parE*	Par E 1232-F	GGC AAT GTG CAG ACC ATC AG	54	265	[68]
	Par E 1498 R	TAC CGA GCT GTT CCT TGT GG			
*qnrA*	QnrAm-F	ATT TCT CAC GCC AGG ATT TG	53	516	[69]
	QnrAm-R	GAT CGG CAA AGG TTA GGT CA			
*qnrB*	QnrBm-F	GAT CGT GAA AGC CAG AAA GG	53	469	[69]
	QnrBm-R	ACG ATG CCT GGT AGT TGT CC			
*qnrS*	QnrSm-F	ACG ACA TTC GTC AAC TGC AA	53	417	[69]
	QnrSm-R	TAA ATT GGC ACC CTG TAG GC			
*qepA*	qepA-F	GCA GGT CCA GCA GCG GGT AG	60	199	[70]
	qepA-R	CTT CCT GCC CGA GTA TCG TG			
*oqxA*	oqxA-F	GAC AGC GTC GCA CAG AAT G	62	339	[71]
	oqxA-R	GGA GAC GAG GTT GGT ATG GA			
*oxqB*	oqxB-F	CGA AGA AAG ACC TCC CTA CCC	62	240	[71]
	oqxB-R	CGC CGC CAA TGA GAT ACA

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
