# Peer review of "Evaluation of Fluoroquinolone Resistance in Clinical Avian Pathogenic Escherichia coli Isolates from Flanders (Belgium)"

_antibiotics, 2020, doi:10.3390/antibiotics9110800_

Round 1

Reviewer 1 Report

Rapid development and selection of resistance to fluoroquinolones is a significant problem. The aim of this study was to investigate the occurrence and mechanisms of antimicrobial resistance against enrofloxacin in avian pathogenic Escherichia coli strains in Flanders, Belgium. 

The study is correctly performed, the methods are adequate and the data are sound.

Minor concerns

The O1, O2, O8, O18 and O78 serotypes are the most frequently detected in clinical cases in poultry. Why only O1, O2 and O78 were tested?.

For MPC determination by agar dilution, concentration of the plates where no visible bacterial growth was detected after 3 consecutive days was designated as the MPC, but some growing colonies on the agar plates were screened for contamination using MALDI-TOF. How many, if any, contaminants were identified, in addtion to the identified E. coli strains?.

Figure 1 "The full lines marked CBS and CBD" should be "The full lines marked CBS and CBR"

One limitation of the study was that other qnr genes (C, D) and the variant of the aminoglycoside acetyltransferase which is responsible for fluoroquinolone resistance (aac(6’)-Ib-cr) were not investigated. It could be possible to characterize some of these genes, or to give a better reason for not doing that?.

Author Response

The O1, O2, O8, O18 and O78 serotypes are the most frequently detected in clinical cases in poultry. Why only O1, O2 and O78 were tested?.

Due to practical constraints, we selected only 3 antisera for serotype determination. According to several studies, O1, O2 and O78 are the three most prevalent serotypes in Western Europe (Ewers et al., 2004). However, the other mentioned serotypes are indeed relevant as well, especially when assessing the global prevalence.  This limitation of the study has been included in the manuscript (lines 229-230).

Ewers, C.; Janßen, T.; Kießling, S.; Philipp, H.C.; Wieler, L.H. Molecular epidemiology of avian pathogenic Escherichia coli (APEC) isolated from colisepticemia in poultry. Vet. Microbiol. 2004, 104, 91–101, doi:10.1016/j.vetmic.2004.09.008.

For MPC determination by agar dilution, concentration of the plates where no visible bacterial growth was detected after 3 consecutive days was designated as the MPC, but some growing colonies on the agar plates were screened for contamination using MALDI-TOF. How many, if any, contaminants were identified, in addtion to the identified E. coli strains?.

We confirm that all the suspected contaminants that were investigated with MALDI-TOF were E. coli strains. No other strains were detected using MALDI-TOF.

Figure 1 "The full lines marked CBS and CBD" should be "The full lines marked CBS and CBR"

This has been adapted in the manuscript.

One limitation of the study was that other qnr genes (C, D) and the variant of the aminoglycoside acetyltransferase which is responsible for fluoroquinolone resistance (aac(6’)-Ib-cr) were not investigated. It could be possible to characterize some of these genes, or to give a better reason for not doing that?.

We thank the reviewer for pointing this out. This limitation is mentioned in the discussion of the manuscript, namely on lines (234-240). The reason for not including these genes was due to practical constrains, only a selection of primers could be acquired, and due to the fact that these plasmid associated genes play a limited role in fluoroquinolone resistance in bacteria originating from poultry.

Reviewer 2 Report

The manuscript entitled "Evaluation of fluoroquinolone resistance in clinical avian pathogenic Escherichia coli isolates from Flanders (Belgium)" details investigation of fluoroquinolone resistance in avian pathogen E. coli isolated from Flanders. 

Comments:

1) What was the reason to evaluate enrofloxacin in this study and not to include other fluoroquinolones that are commoly used in veterinary (e.g.: marbofloxacin). In my opinion in parallel marbofloxacin and enrofloxacin should be evaluated and selection power (MPC values ) of these agents could be evaluated. 

2) In the "Introduction" it should be mentioned that qnrE is also a PMQR determinant: Reference article: Albornoz et al: qnrE1, a Member of a New Family of Plasmid-Located Quinolone Resistance Genes, Originated from the Chromosome of Enterobacter Species,  Antimicrob Agents Chemother. 2017;61(5):e02555-16. doi: 10.1128/AAC.02555-16.

3) In the introduction "aac(6’)- ib - cr" is written, however proper form is: aac(6’)- Ib - cr. In other parts of the text it is well written. 

4) All bacterial names E. coli, Escherichia coli, and all gene names must be written in italics form : gyrA, parC etc.

Author Response

Comments:

1) What was the reason to evaluate enrofloxacin in this study and not to include other fluoroquinolones that are commoly used in veterinary (e.g.: marbofloxacin). In my opinion in parallel marbofloxacin and enrofloxacin should be evaluated and selection power (MPC values ) of these agents could be evaluated.

Thank you for your comment. Only enrofloxacin (second generation fluoroquinolone) is registered for use in Belgium. This is similar for other Western European countries where marbofloxacin (e.g. Marbocyl®, Marboflox®) is registered for use in companion animals, pigs and cattle but not in poultry. Therefore it was not included in the study.

2) In the "Introduction" it should be mentioned that qnrE is also a PMQR determinant: Reference article: Albornoz et al: qnrE1, a Member of a New Family of Plasmid-Located Quinolone Resistance Genes, Originated from the Chromosome of Enterobacter Species,  Antimicrob Agents Chemother. 2017;61(5):e02555-16. doi: 10.1128/AAC.02555-16.

The qnrE gene as an additional PMQR determinant and the reference has been added to the manuscript.

3) In the introduction "aac(6’)- ib - cr" is written, however proper form is: aac(6’)- Ib - cr. In other parts of the text it is well written.

This has been adapted in the manuscript.

4) All bacterial names E. coli, Escherichia coli, and all gene names must be written in italics form : gyrA, parC etc.

This has been adapted in the manuscript accordingly.

Author Response

Below are my comments and suggestions for improving your manuscript. Comments and Suggestions for Authors

Line 39 (Keywords): the keywords are repetitive and should never be the same as indicated in the title. I suggest modifying them above all to increase the availability of the article. For example, I suggest you insert “MIC” and “Antimicrobial resistance”.

Thank you for the advice. We have inserted MIC and Antimicrobial resistance into the keywords and dropped PMQR and QRDR. In addition, avian pathogenic Escherichia coli has been rewritten as APEC and mutant prevention concentration as MPC.

Line 76: I suggest you remove this sentence or to integrate it into the aim of the study.

Per the reviewer’s request, this line has been deleted from the manuscript.

Lines 99-105: The purpose of the study is summarized in two lines. The following period represents a synthesis of the work done. I think it is useful to remove 100 to 105 and I suggest to insert the some sentences that can explain the impact the study could have on the scientific community.

The paragraph partly aims to explain why we chose only 3 antisera  (lines 99-100) per the reviewer’s request (see comment below). If agreed by the reviewer, the authors would like to keep this paragraph for fluidity and continuity. However, based on the reviewer’s advice, additional information was included at the end of the paragraph to explain the impact of this study on the scientific community (lines 104 – 107).

Line 109 (Results): Perhaps it would have been interesting to know if there were O8 and O18 Strains, explaining why they have not been studied or researched. Please clarify.

Due to practical constraints, we selected only 3 antisera for serotype determination. According to several studies, O1, O2 and O78 are the three most prevalent serotypes in Western Europe (Ewers et al., 2004). However, the other mentioned serotypes are indeed relevant as well, especially when assessing the global prevalence.  This has been included in the manuscript.

Ewers, C.; Janßen, T.; Kießling, S.; Philipp, H.C.; Wieler, L.H. Molecular epidemiology of avian pathogenic Escherichia coli (APEC) isolated from colisepticemia in poultry. Vet. Microbiol. 2004, 104, 91–101, doi:10.1016/j.vetmic.2004.09.008.

Line 262: As above, why were only these three O-antigens investigated?

We kindly refer to the answer given to the previous question.

Lines 262-266: I do not question the reliability of the agglutination test, but I believe that the interpretation is somewhat subjective. How did you go about avoiding this potential misinterpretation?

It is very true that the interpretation of agglutination reaction is subjective. In order to avoid misinterpretation, the results were also checked with an experienced laboratory technician and only if consensus was reached on the interpretation of the result, it was reported.

Line 173 (Discussion): I suggest that the authors structure a discussion with a subsection for the limitations of the study, including any shortcomings of the study.

Per the reviewer’s request, the limitations of the study are summarized under a new paragraph in the discussion (lines 229-240).

Lines 238-251: This long period represents a full-fledged conclusion, which should be placed in a separate section "Conclusions" which, as reported in the Instructions for the Authors, is not mandatory but also represents the final part of the study on focusing attention of non-expert readers.

Per the reviewer’s request, a new subsection “conclusions” was added containing lines 238-251.

References

The references must be partially changed, since they were not written following the “instructions for authors" of the journal (i.e. font and text formatting). 

The references have been  updated according to the guidelines of the journal (font and text format).